# Ligand-induced conformational changes in the β1-adrenergic receptor revealed by hydrogen-deuterium exchange mass spectrometry

Joanna Toporowska[1], Parth Kapoor[2], Maria Musgaard [2], Karolina Gherbi[2], Kathy Sengmany[2], Feng Qu [2], Mark Soave[2], Hsin-Yung Yen[2], Kjetil Hansen[1], Ali Jazayeri[2], Jonathan T. S. Hopper [2] ✉ & Argyris Politis [1,3,4] ✉

G Protein Coupled Receptors (GPCRs) constitute the largest family of signalling proteins responsible for translating extracellular stimuli into intracellular functions. They play crucial roles in numerous physiological processes and are major targets for drug discovery. Dysregulation of GPCRs is implicated in various diseases, making understanding their structural dynamics critical for therapeutic development. Here, we use Hydrogen Deuterium Exchange Mass Spectrometry (HDX-MS) to explore the structural dynamics of the *turkey* β1-adrenergic receptor (tβ1AR) bound with nine different ligands, including agonists, partial agonists, and antagonists. We find that these ligands induce distinct dynamic patterns across the receptor, which can be grouped by compound modality. Notably, full agonist binding destabilises the intracellular loop 1 (ICL1), while antagonist binding stabilises it, highlighting ICL1's role in G protein recruitment. Our findings indicate that the conserved L72 residue in ICL1 is crucial for maintaining receptor structural integrity and stabilising the GDP-bound state. Overall, our results provide a platform for determining drug modality and highlight how HDX-MS can be used to dissect receptor ligand interaction properties and GPCR mechanism.

G protein coupled receptors (GPCRs), a superfamily of more than 800 unique members, and the largest family of receptor proteins in mammals, are responsible for a diverse range of physiological responses[1,2]. They share a common structural arrangement via seven transmembrane (7TM) helices that are bound by three intracellular (ICL) and three extracellular (ECL) loops. GPCRs respond to various ECL stimuli, including hormones, metabolites and neurotransmitters[1]. Ligand binding to GPCRs serves to induce, or stabilise, distinct conformational arrangements that define the ICL physiological response. Canonical agonist binding,

for example, results in GPCR conformations that selectively allow the receptor to recruit and activate specific heterotrimeric G proteins, resulting in the generation of second messenger molecules. Additionally, some ligands induce a functionally selective (or biased) response, driving signalling through either specific G proteins or through other transducers such as β-arrestins. These differences in signalling pathways, which essentially define the physiological action of a particular ligand, are driven by allosteric conformational changes of the receptor. This highlights the challenges in future GPCR drug discovery, where early knowledge

[1]King's College London, London, UK. [2]OMass Therapeutics, Oxford, UK. [3]Faculty of Biology, Medicine and Health, School of Biological Sciences, The University of Manchester, Manchester, UK. [4]Manchester Institute of Biotechnology, The University of Manchester, Manchester, UK. ✉ e-mail: Jonathan.hopper@omass.com; argyris.politis@manchester.ac.uk

of the ligand-induced conformational dynamics of the receptor can greatly improve the identification of early hits with the correct pharmacological response, as well as drive correct decisions during the structure-activity relationship (SAR) development[3].

The β1-adrenergic receptor (β1AR) belongs to the class A, or rhodopsin-like, subfamily of GPCRs along with β2AR and β3AR, which share more than 50% of sequence similarity[4]. β1AR is abundantly expressed in cardiac tissue and is critical for the physiological regulation of heart rate. Antagonists of β1AR are often referred to as β-blockers and inhibit binding of endogenous ligands norepinephrine and adrenaline to the catecholamine pocket, ultimately preventing receptor activation and resulting in a reduction in heart rate[5,6]. Structurally most of the transmembrane region is highly conserved among all GPCRs. The remarkable differences in the sequence are mostly present in the ICL and ECL regions which allows the wide range of ligands to bind to the GPCR orthosteric site and regulate receptor activity[1]. Mechanisms and structural transitions behind receptor activation are still unclear.

Hydrogen deuterium exchange mass spectrometry (HDX-MS) has emerged as powerful tool for monitoring the exchange of backbone hydrogen atoms for heavier deuterium in bulk $D_2O$ over different time intervals[7-10]. The rate of exchange depends on the local environment around a given backbone hydrogen atom, as well as the hydrogen bonding and accessibility of the local segment in which the specific hydrogen atom is located. Thus, measuring the rate of exchange can reveal details about local structural changes and dynamics. Here, deuterium uptake across the protein is determined by proteolytically digesting the protein and subsequently using MS to measure the mass changes present across the identified peptides[10,11]. Recent workflows developed in our group and others have enabled experimental measurements of integral membrane proteins which achieve high sequence coverage[12-14] and allow a more comprehensive analysis of dynamic conformation changes in these targets. HDX-MS offers advantages over traditional approaches; it tolerates heterogenous environments (e.g. lipid nanodiscs), has low sample requirements and there is no need for chemical probes or bio-orthogonal labels[10,15]. Despite advances and early successes[16,17], interrogating the conformational dynamics of GPCRs remain a challenging task, primarily owing to difficulties in solubilising functionally competent preparations of these important biomolecules. To overcome that issue and efficiently solubilise membrane proteins, many different solutions or methods have been proposed including surfactants (e.g. detergents) or nanodiscs (lipid-based environment) and styrene-maleic acid-lipid particle technology[12,18]. They provide a protective layer surrounding the transmembrane part of the protein. However, the resulting heterogeneous mixtures may complicate the experiments, leading to extra steps in the optimisation protocol (e.g. removal of lipids)[10]. Here we have successfully optimised the use of HDX-MS to be able to interrogate the structural dynamics of tβ1AR[19].

Here, we deploy differential HDX-MS to monitor the global conformational dynamics of tβ1AR in response to a diverse range of ligands that are linked to defined pharmacological responses upon binding to the receptor. Our findings illustrate distinct dynamic signatures across the receptor upon full, partial agonist, and antagonist binding. Interestingly, opposing structural effects (deprotection vs. protection) are consistently found in the ICL loop 1 (ICL1), which appears to highlight a structurally essential component in driving receptor activation vs. deactivation. This prompts us to further explore the structure and function of ICL1 using site-directed mutagenesis in combination with molecular dynamics simulations and cell-based functional assays. We provide a peptide-level characterisation of the loop and reveal the impact of ICL1 on downstream signalling in a class A receptor.

## Results

### Optimising conditions to achieve high peptide coverage in HDX-MS

To date, most of the studies of GPCRs by HDX-MS have been frustrated by low to medium sequence coverage. This is primarily due to low yield of purification and inherent instabilities of GPCRs as well as the lack of appropriate methodologies to maximise the peptide coverage in HDX-MS experiments. As such, we began our investigations by optimising conditions to achieve high sequence coverage for tβ1AR (Supplementary Methods). Initially we screened against various quench buffer compositions as well as different pepsin columns (Supplementary Fig. 1). Optimal sequence coverage was achieved using a self-packed pepsin column with immobilised pepsin agarose, which allowed us to achieve a sequence coverage of 75%. Moreover, we observed that increasing DDM (n-dodecyl-β-D-maltopyranoside) concentration in quench buffer from 0.01% to 0.1%, ~10x the documented critical micelle concentration, resulted in significant increase (~20%) in sequence coverage. It has also been reported that tβ1AR contains disulfide bonds which can negatively impact on the efficiency of enzyme digestion[20]. Therefore, to achieve optimal sequence coverage, we maintained a tris(2-carboxyethyl)phosphine hydrochloride (TCEP) concentration of 100 mM to reduce disulfides. Finally, we optimised the digestion step by utilising a dual protease type XIII/pepsin column. The protease type XIII (from *Aspergillus saitoi*) is a non-specific enzyme, and its cleavage sites are complementary to pepsin[21]. Utilising this protease column led to a significant increase in sequence coverage of up to 94%, and redundancy of 8.27 (Supplementary Fig. 2). With the optimal quench composition and digestion efficiency established, we progressed to full HDX-MS characterisation of drug binding.

### HDX fingerprint of tβ1AR bound to antagonist

Initially, we explored the conformational dynamics of tβ1AR upon binding to three antagonists, namely cyanopindolol, carazolol, and carvedilol. Cyanopindolol and carazolol are high-affinity antagonists for both β1- and β2-adrenergic receptors[22]. Carvedilol is a specific β1AR biased ligand that preferentially stimulates the β-arrestin signalling cascade, while displaying antagonistic properties towards G protein[23].

We conducted HDX-MS experiments across four-time points (15 s, 2 min, 30 min and 120 min) and used excess of ligand concentrations to achieve >98% ligand binding occupancy based on the reported affinity constants (Supplementary Table 2). We observed that various regions of the receptor are protected from HDX for all three ligand-receptor complexes (Fig. 1A–C). Such behaviour is expected and is related to the increased stabilisation of its secondary structure and reduced receptor activity. The protection observed in TM5 is induced by binding of the ligand in the catecholamine binding pocket located between ECL parts of TM3, 4, 5, 6 and 7. Previous crystallographic studies of antagonists/inverse agonist structures illustrated that binding is localised to the TM7 and TM5 interface, involving key interactions with residues Ser211, Ser212 and Ser215[24,25]. Our data further revealed additional protection on the tip of TM4 and ECL2. This region contains a short α-helical segment that has previously been shown to be a component of the ligand binding site, forming the so-called lid which enables access to the orthosteric site[1]. Recent studies on β2AR and β1AR receptors, focusing on the influence of ligand flexibility and the entropic component on catecholamine binding, have unveiled significant insights. These investigations have revealed that the shape and stability of the orthosteric pocket can be influenced by surrounding residues, leading to notable variations in ligand affinity. Specific residues within ECL loop 2 (ECL2) have been identified as key modulators, capable of reshaping the catecholamine binding pocket and distinguishing between different ligand-bound ensembles[26]. Consistent with this finding, several studies indicate that ECL2 is crucial for ligand specificity and determines the affinity of ligands for the receptor. Research suggests that the binding pathway

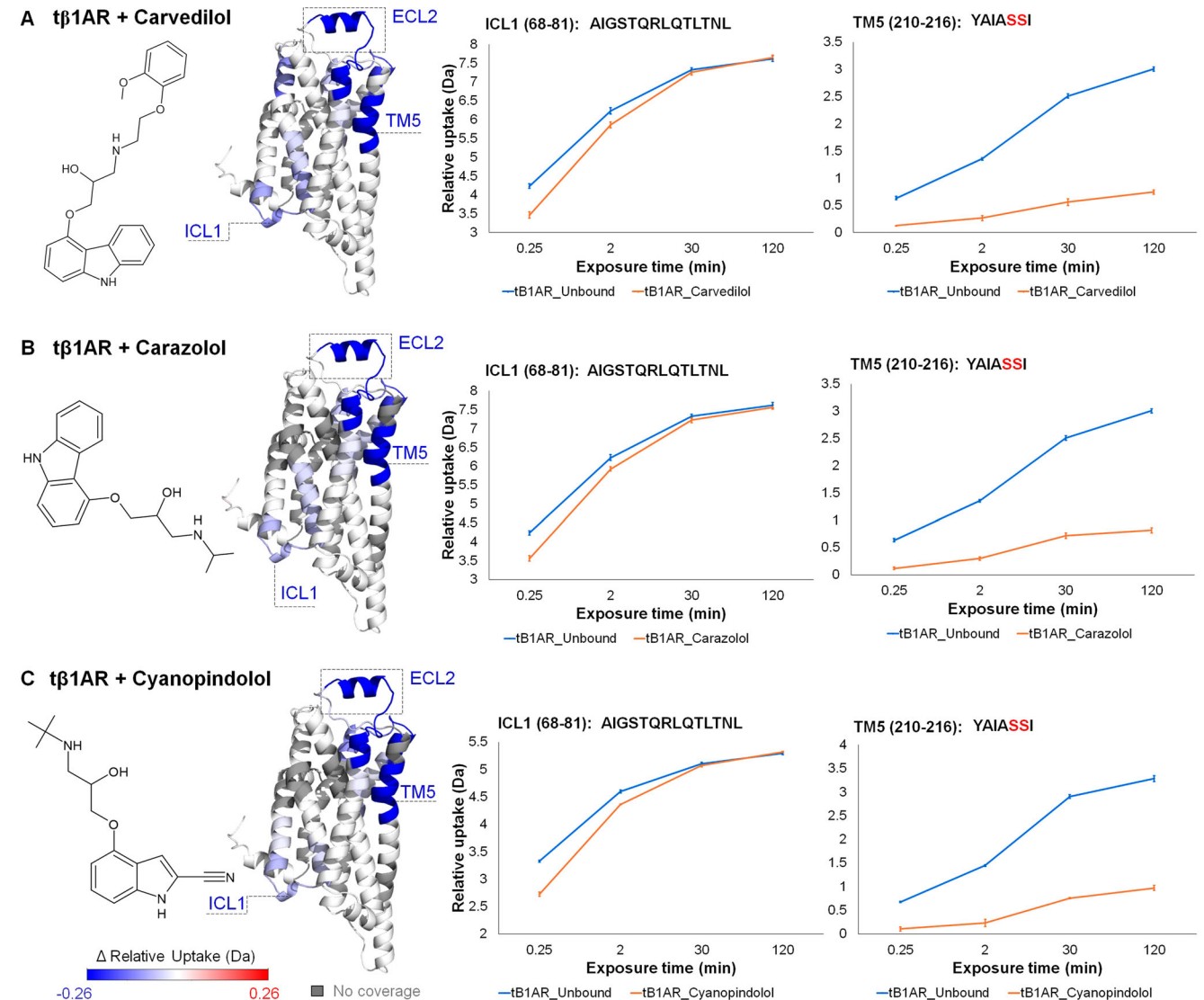

**Fig. 1 | Conformational dynamics of tβ1AR upon binding to antagonist.**
**A** Chemical structure of antagonist carvedilol, ΔHDX-MS results for tβ1AR + carvedilol vs tβ1AR mapped on β1AR structure, deuterium uptake plots for ICL1 and TM5. **B** Chemical structure of antagonist carazolol, ΔHDX-MS results for tβ1AR + carazolol vs tβ1AR. **C** Chemical structure of antagonist cyanopindolol, ΔHDX-MS results for tβ1AR + cyanopindolol vs tβ1AR. Results are mapped on tβ1AR structure (modelled using PDB structures; 2VT4 chain A and 6IBL chain A). Ser residues,

crucial for binding the ligands into catecholamine binding pocket of the receptor, are highlighted in red. Relative deuterium uptake shows cumulative data across all time points for every receptor-ligand complex. Data are presented as mean values ± SD, where error bars represent the standard deviation at each time point. Each measurement is based on three technical replicates ($n = 3$). Source data are provided as a Source Data file.

of different ligands consists of two major steps: the ligand first passes between ECL2 and ECL3, then traverses a narrow passage between ECL2 and TM5, TM6, and TM7 before reaching the main binding pocket[27]. These findings conclude that ECL2 plays key roles in the early stages of ligand recognition and in determining binding kinetics[4,28]. It is worth noting that protection on the tip of TM4 spans across all time points with similar difference in deuterium uptake (-0.6 Da), whereas on ECL2 the effect manifests only at 15 s and 2 min time points. We attribute the latter to a higher flexibility of the loop region that forms transient interactions in the presence of antagonists. Our data also demonstrated that binding of antagonists induces stabilising effects on ECL3 and the tips of TM6 and TM7, which increase with time (Supplementary Fig. 17). Computational studies have previously predicted that a secondary, transient binding site is located on ECL2, ECL3 and TM7, which may explain protection in these regions[28].

In addition to protection on the ECL side of β1AR, we unexpectedly observed repeated stabilising effects on ICL1 for all tested

antagonists, which to our knowledge has not been reported previously. It is interesting to note that carvedilol shows unique properties by inducing slightly different dynamics in comparison to other antagonists tested. Specifically, it displays additional protection on the kink of TM2 that is located between TM1 and 7 (Supplementary Fig. 17). This can be explained by its larger chemical structure containing the additional 3,4-dimethoxyphenethyl group, that extends beyond the catecholamine binding pocket in the direction of ECL ends of TM1, 2 and 3. As mentioned above, residues located in this secondary binding cavity could be responsible for receptor selectivity of bigger ligands as well as to play a role in biased signalling[28]. Studies conducted on the angiotensin II receptor type 1 with a biased ligand demonstrate that its ability to signal through the β-arrestin pathway highly depends on the conserved proline located on the end of TM2[29]. More importantly, our data demonstrate that all the antagonists investigated result in a reproducible global reduction in receptor dynamics. This suggests that compounds with common pharmacological modality may be

identified in the absence of cell-based assays or downstream binding partners, simply by monitoring the changes in conformational dynamics induced by ligand binding. To test this hypothesis further, we expanded these studies to investigate receptor dynamics in the presence of agonists.

## HDX fingerprint of tβ1AR bound to agonist

Having established the conformational fingerprint in tβ1AR upon antagonist binding, we set out to probe the conformational effects of two agonists, isoprenaline and norepinephrine, an endogenous ligand of β1AR. Previous structural studies illustrated that whilst agonists bind to the same β1AR catecholamine binding pocket as antagonists, they undergo additional rotamer conformational changes in Ser212 and Ser215[28,30]. Similar to antagonist binding, agonist binding induced protection on TM5, TM4-ECL2 and TM6-ECL3-TM7, which sits directly within the orthosteric site and contains residues essential for binding (Fig. 2A, B). In contrast to antagonist binding, we observed deprotection, consistent with increased receptor dynamics, in multiple regions of the receptor including ICL1, ECL1, ICL2, ICL3, TM6 and helix 8 (H8) (Supplementary Figs. 18, 19). Despite general consistency in deprotection patterns for both agonists, we observed distinct differences between the two agonists, including protection on the tip of TM4, and deprotection on ICL3 observed for isoprenaline but not for norepinephrine. The lack of protection on TM4 for norepinephrine may suggest a slightly different docking mechanism for this ligand in receptor's secondary binding site. It is hypothesised that ligands with lower agonist affinity exhibit greater mobility when entering the receptor's secondary binding site[27]. Norepinephrine has lower affinity than isoprenaline which could affect the ligand's pose during its entry to the binding pocket. It is well established that in all GPCRs, ICL loops play important roles in receptor activation, selectivity, and specificity[4]. Previous studies have reported that coupling of a GPCR to G protein involves interactions through the cytoplasmic loops, with ICL loop 3

(ICL3) playing a significant role. ICL3 might not only contribute to determining specificity between different G proteins but also tunes signalling specificity by inhibiting receptor coupling to G protein subtypes that weakly couple to the receptor[19,31]. Class A GPCRs couple to G protein at the cytoplasmic side mostly involving the end of TM3/ICL2, TM6 and TM5/ICL3[32]. Recent native MS studies state that iso-prenaline induced β1AR complex formation with miniG$_i$, whereas no complex formation was observed with norepinephrine[19]. Consistent with these findings, we have observed increased deuterium uptake on ICL3 (Supplementary Fig. 18) for isoprenaline only, which highlights its ability to act as a biased agonist. This underscores the crucial role of ICL3 in defining specificity for distinct G proteins and facilitating biased signalling pathways, where different ligands binding to the same receptor can selectively activate specific downstream signalling cascades, leading to distinct physiological responses[19].

Recent NMR study on β2AR-Gs and β2AR-Gi complexes high-lighted differences in conformational changes in ICL2 between the two complexes, suggesting that ICL2 may also play a role in G-protein coupling selectivity[33]. Additionally, it has been proposed that ICL2 acts as a switch, facilitating G protein binding and interacting with the receptor's conserved DRY motif located on TM3[4]. Increased deuterium uptake for both loops suggests that they play fundamental roles in recruitment of G proteins and ultimately in receptor activation. Our data also reveals deprotection on TM6 and the cytoplasmic end of TM5 for both ligand-receptor complexes with similar level of deuterium incorporation (-0.6 Da for TM6). This effect supports a large outward movement of TM6 and TM5 upon agonist binding, confirming that those helices undergo significant conformational changes during receptor activation[34]. Notably, the effect on TM6 is only observed for the peptides containing three non-polar amino acid residues; two Isoleucine (I269, I270) and Methionine (M271) located closely to the core of TM6. This observation likely underlines the importance of those residues in so-called pivotal movement of TM6 around its centre,

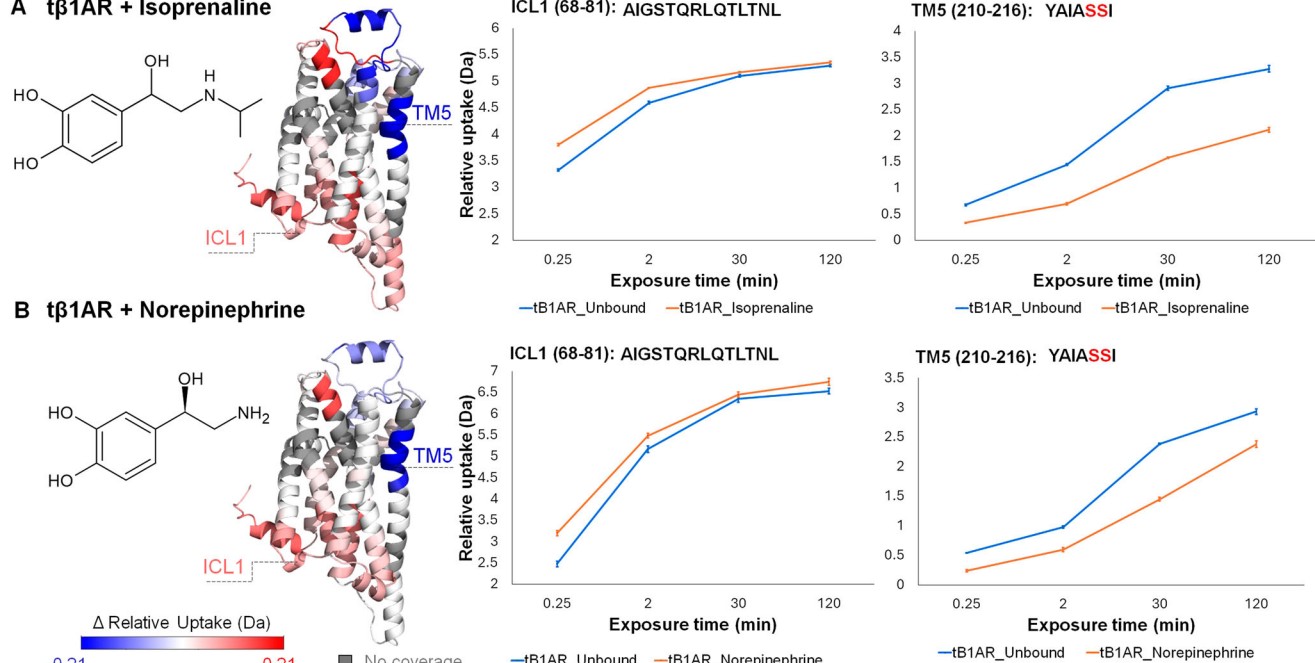

**Fig. 2 | Conformational dynamics of tβ1AR upon binding to agonist. A** Chemical structure of agonist isoprenaline, comparative HDX-MS results for tβ1AR + isoprenaline vs tβ1AR, deuterium uptake plots for ICL1 and TM5. **B** Chemical structure of endogenous agonist norepinephrine, ΔHDX-MS results for tβ1AR + norepinephrine vs tβ1AR, deuterium uptake plots for ICL1 and TM5. Results are mapped on tβ1AR structure (modelled using PDB structures; 2VT4 chain A and 6IBL

chain A). Ser residues, crucial for binding the ligands into catecholamine binding pocket of the receptor, are highlighted in red. Relative deuterium uptake shows cumulative data across all time points for every receptor-ligand complex. Data are presented as mean values ± SD, where error bars represent the standard deviation at each time point. Each measurement is based on three technical replicates ($n = 3$). Source data are provided as a Source Data file.

where the ICL side of the helix moves outwards, and the ECL side moves inwards closer to the binding pocket[35]. Unexpectedly, we found that a unique characteristic of both agonists was their ability to induce increased dynamics on ICL1, in contrast to antagonists which lead to decreased dynamics on ICL1 of tβ1AR.

## HDX fingerprint of tβ1AR bound to partial agonist

Next, we turned our attention to two partial agonists for β1AR and β2AR, dobutamine and salbutamol, respectively. For both partial agonist-tβ1AR complexes we measured similar reduced dynamics on the ECL side of tβ1AR (ECL2, TM5, TM6-ECL3-TM7) as for agonist and antagonist (Fig. 3A, B), driven by the protection afforded due to ligand binding. Interestingly, on ECL2 we observed significantly more protection for antagonist than for agonist or partial agonist (-1.2 Da, 0.5 Da, 0.4 Da, respectively). On TM5 we noted various deuterium uptake differences between the ligands, with all tested antagonists inducing protection of ~2 Da, compared to agonists (isoprenaline -1.3 Da and norepinephrine 0.5 Da) and partial agonists (dobutamine 1.9 Da and salbutamol 0.7 Da). It is interesting to speculate that such variety in deuterium uptakes between different ligands is potentially linked to the affinity of the compounds to the receptor (Supplementary Table 2). Higher affinity ligands generally stabilise receptor conformations more effectively, leading to reduced hydrogen-deuterium exchange. This stabilisation effect occurs even under comparable % bound conditions because high-affinity ligands tend to have slower off rates, maintaining their interaction with the receptor for longer periods and inducing greater protection. Thus, the observed differences in deuterium uptake likely reflect the varying abilities of ligands to stabilise the receptor, influenced by their relative on/off rates and binding strengths. Protection on TM6-ECL3-TM7 does not vary much between

antagonist, agonist, and partial agonist (averaged values for all ligands: -0.6 Da, 0.5 Da, 0.4 Da uptake difference, respectively). For dobutamine, we further observed additional stabilisation induced upon ligand binding on the tip of TM2 of tβ1AR (Supplementary Fig. 19). Dobutamine is a considerably larger ligand, and once bound to the receptor would extend beyond the catecholamine binding pocket, causing protection on TM2; this is line to our data for the biased agonist carvedilol. TM2 has also been recognised by NMR studies as an important location indicative of distinct receptor conformations, including carvedilol-bound state[36]. Similar to norepinephrine, salbutamol did not induce protection on the ECL tip of TM4, whereas all other tested ligands did. Most likely, those observations highlight the different modes of ligand binding in the orthosteric binding pocket and their ability to stabilise the ECL site of the receptor. Notably, the pharmacophore moiety of salbutamol docks in the catecholamine binding pocket in the same manner as endogenous agonist norepinephrine[37,38], which possibly explains the lack of protection on the tip of TM4 for both ligand-receptor complexes.

The major differences between those two partial agonist-receptor complexes were noted on the cytoplasmic site of tβ1AR. Salbutamol-bound tβ1AR is considerably more dynamic than dobutamine-bound tβ1AR (deprotection only on H8) inducing additional deprotection on ICL2 and TM6, however, still less than the agonist-receptor complex, which displays the most dynamic state of the receptor. Deprotection on TM6, with the average statistically significant difference in deuterium uptake (120 min time point) of -0.5 Da, -0.6 Da and -0.8 Da for salbutamol, norepinephrine and isoprenaline respectively, indicates smaller outward movement of TM6 for salbutamol-receptor complex. The number of peptides displaying the effect on TM6 was also significantly lower for salbutamol (Supplementary Fig. 7). This is

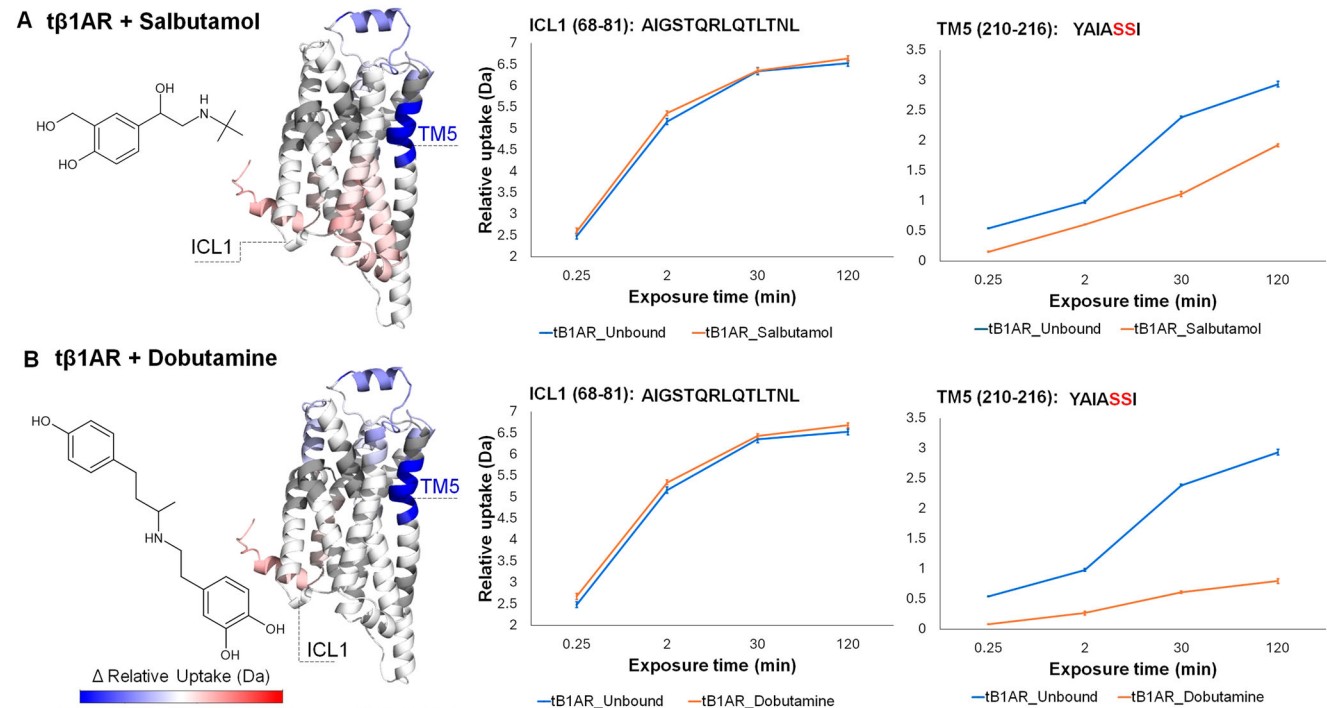

**Fig. 3 | Conformational dynamics of tβ1AR upon binding to partial agonist.**
**A** Chemical structure of partial agonist salbutamol, ΔHDX-MS results for tβ1AR + salbutamol vs tβ1AR mapped on β1AR structure, deuterium uptake plots for ICL1 and TM5. **B** Chemical structure of partial agonist dobutamine, ΔHDX-MS results for tβ1AR + dobutamine vs tβ1AR mapped on β1AR structure, deuterium uptake plots for ICL1 and TM5. Results are mapped on tβ1AR structure (modelled using PDB

structures; 2VT4 chain A and 6IBL chain A). Ser residues, crucial for binding the ligands into catecholamine binding pocket of the receptor, are highlighted in red. Relative deuterium uptake shows cumulative data across all time points for every receptor-ligand complex. Data are presented as mean values ± SD, where error bars represent the standard deviation at each time point. Each measurement is based on three technical replicates (n = 3). Source data are provided as a Source Data file.

potentially indicative of an intermediate active state of the receptor related to salbutamol's partial agonism[37]. Observed differences between dobutamine and salbutamol might be linked to variation in affinity between those two partial agonists (salbutamol: $k_d = 20.89$ μM, dobutamine:$k_d = 5.88$ μM)[39], and they both could propagate slightly different dynamics across the receptor spanning from binding pocket to cytoplasmic site of the receptor. Crystallographic studies of the β1AR in complex with agonist and partial agonist reveal that the difference in binding between those two classes of drugs is a reduction in the number of hydrogen bonds between the ligand and the residues located in catecholamine binding pocket, whereas motifs in the ICL site of the receptor were almost the same[30]. In addition, partial agonists do not alter the conformation of Ser215[30]. It is worth noting that there were no effects observed on ICL1 for both partial agonists. Our data show clear differences between the receptor dynamics of antagonist-, full agonist-, and partial agonist- bound tβ1AR, information that is ultimately inaccessible in structure determination through x-ray crystallography or cryo-electron microscopy. Such dynamics provide further insight in understanding receptor movements and responses to drugs, therefore illustrating the power of combining multiple complementary approaches for understanding GPCR function.

## The role of ICL1 in receptor activation

An unexpected yet consistent finding of our investigations was the altered dynamics observed in ICL1 in response to various ligands bound to tβ1AR. To explore the role of ICL1 in receptor activation or G protein recruitment, we extended our HDX-MS studies to include two additional ligands, derivatives of full agonist isoprenaline, namely colterol hydrochloride and orciprenaline. Whereas the first can maintain a level of coupling activity similar to isoprenaline, orciprenaline is characterised by a significantly impaired ability to induce β1AR-G protein recruitment[19]. Interestingly, while our data revealed increased dynamics of ICL1 in the presence of colterol hydrochloride, similar to the level observed for isoprenaline, no significant effect was observed in the presence of orciprenaline (Fig. 4A–C). Taken together, the increased dynamics in ICL1 are only established upon full activation of the receptor, and not observable in the presence of partial agonists. Further, the result with orciprenaline above suggests that ICL1 may play an essential role in allowing agonists to transduce signalling down the G protein pathway.

To investigate the impact of ICL1 on receptor function, we performed mutational analysis in the ICL1 region of tβ1AR. The central part of ICL1 in tβ1AR is composed of four residues; Q70, R71, L72 and

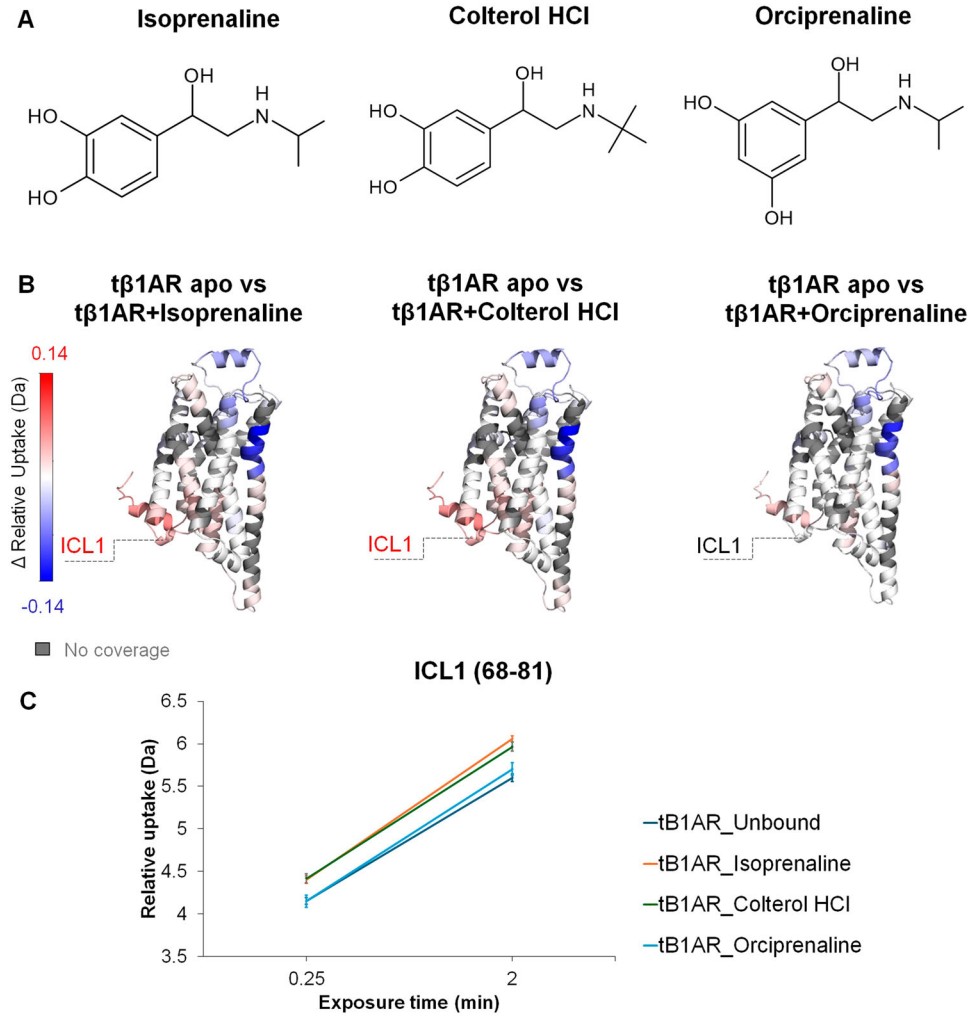

**Fig. 4 | ΔHDX analysis of tβ1AR apo vs tβ1AR bound to isoprenaline, colterol HCl and orciprenaline.** A Chemical structure of tested ligands, full agonist isoprenaline and its derivatives colterol hydrochloride and orciprenaline. **B** Regions showing significant difference in HDX for tβ1AR with isoprenaline derivatives mapped onto β1AR structure (modelled using PDB structures; 2VT4 chain A and 6IBL chain A). Relative deuterium uptake shows cumulative data across all time points for every receptor-ligand complex. **C** Uptake plot for ICL1 displaying differences in relative deuterium uptake (Da) between tβ1AR Unbound (apo) and tβ1AR + isoprenaline, tβ1AR+colterol HCl, and tβ1AR + orciprenaline, respectively. Data are presented as mean values ± SD, where error bars represent the standard deviation at each time point. Each measurement is based on three technical replicates ($n = 3$). Source data are provided as a Source Data file.

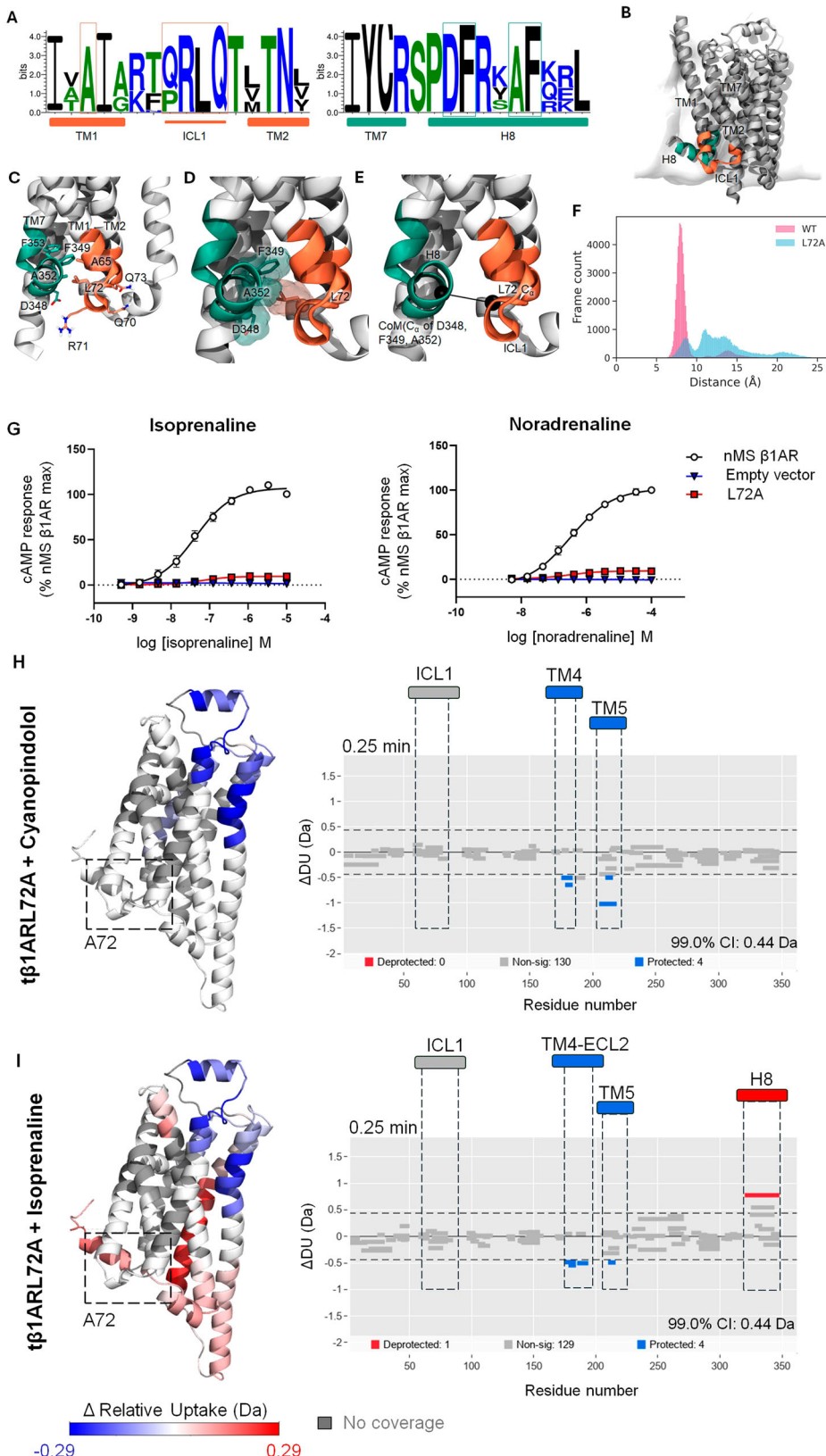

Q73. Initially, we generated a sequence logo (Fig. 5A) based on multiple sequence alignments for the top 1000 hits from a Blast search on the wild-type *turkey* β1A sequence. The sequence logo illustrated that residues R71, L72 and Q73 (β1AR numbering) are well conserved. It is evident from the structure of tβ1AR that ICL1 is in close proximity of the horizontal H8 (Fig. 5B). Inspecting potential

direct interactions between ICL1 and H8 (Fig. 5C) revealed that the lesser-conserved Q70 but also the more conserved Q73 point away from H8 and do not seem to form direct interactions with other structural elements of the protein. R71 of ICL1 could potentially form a salt bridge to D348 of H8, although their arrangement does not indicate a strong salt bridge. L72, on the other hand, seems to

**Fig. 5 | Design, dynamics and functional consequence of β1AR L72A. A** Sequence logos for the 1000 top sequence from a blast search on the *turkey* WT β1AR sequence, showing the cytoplasmic end of TM1 and TM2 along with the connecting ICL1, and cytoplasmic end of TM7 with the H8 helix. Secondary structure elements are highlighted with the solid-coloured boxes below the plots. **B** Overall structure of β1AR (PDB: 4AMJ)[40] illustrated in cartoon, placed in a POPC membrane model. ICL1 and H8 are highlighted using the same colours as the secondary structure bars in panel A. **C** An enlargement of the ICL1-H8 region with potentially important residues shown in licorice. The included residues are highlighted on panel A in thin boxes. **D** Same view as **C** but highlighting the packing of L72 against D348, F349 and A352. The $C_\alpha$ atoms are shown in vdW representation, and the side chains are shown with licorice as well as transparent surface. **E** Definition of the ICL1-H8 distance which is measured between the two black spheres defined by the $C_\alpha$ carbon atom of L72A and the centre of mass (CoM) of the $C_\alpha$ carbon atoms of D348, F349 and A352, respectively. **F** Distribution of ICL1-H8 distances as measured for a total of 1.2 μs of MD simulation for each of the β1AR wild type (pink) and the L72A

mutant (blue). **G** Isoprenaline and noradrenaline concentration response curves for cAMP accumulation determined in CHO cells transiently transfected with either empty vector, wild type (nMS) or L72A tβ1AR. Data are expressed as mean ± SEM of *n* independent experiments performed in duplicate; nMS β1AR (isoprenaline *n* = 5, noradrenaline *n* = 4); empty vector (isoprenaline *n* = 11, noradrenaline *n* = 7); L72A β1AR (isoprenaline *n* = 8, noradrenaline *n* = 7). Error bars not shown lie within the dimensions of the symbol. **H** Regions showing significant difference in HDX for tβ1AR L72A with agonist isoprenaline and Woods plot displaying statistically significant peptides for 0.25 min time point. **I** Regions showing significant difference in HDX for tβ1AR L72A with antagonist cyanopindolol and Woods plot displaying statistically significant peptides for 0.25 min time point. Results are mapped onto β1AR structure (modelled by the use of PDB structures; 2VT4 chain A and 6IBL chain A). Woods plots were generated by Deuteros 2.0[34]. Relative deuterium uptake shows cumulative data across all time points for every receptor-ligand complex. Source data are provided as a Source Data file.

potentially pack into a small hydrophobic pocket formed especially by the $C_\alpha$ of D348 along with F349 and A352 of H8 (Fig. 5D) but also with potential contributions from A65 in TM1 and F353 in H8. Especially F349 and F352 are well-conserved. We predicted that these interactions may be key for controlling the dynamics of ICL1. To disrupt this packing of ICL1 and H8 we mutated L72 to alanine. MD simulations were performed to assess the ICL1-H8 packing in the apo state of both the tβ1AR *turkey* β1AR (modelled as PDB ID:4AMJ[40]) and a computationally constructed L72A mutant. Measuring the distance between ICL1 and H8 (as defined in Fig. 5E) during simulations predict the ICL1-H8 packing to be relatively stable in the tβ1AR apo state, but heavily disrupted in the L72A mutant, showing broader distribution of the distance for the mutant (Fig. 5F).

We designed a mutated construct to allow us to examine the effects of the L72 residue in further experiments. We first determined the functional consequence of these mutations in Chinese hamster ovary (CHO) cells transiently transfected with tβ1AR and L72A tβ1AR by measuring cAMP production in response to isoprenaline and norepinephrine. In support of our predictions, the L72A variant showed a significantly impaired response to agonist, almost completely abolishing the resulting cAMP levels relative to parent tβ1AR for both tested agonists (Fig. 5G). Receptor expression in membrane preparations of CHO cells transiently transfected with tβ1AR and L72A β1AR was determined using Western blot analysis, which confirmed that the L72A receptor was successfully expressed and not misfolded, but that lower expression levels were observed for the mutant compared to the parent receptor (Supplementary Data 1). Taken together, these data suggest a potential key role of ICL1, and specifically the L72 residue, in propagating GPCR signalling, however, we cannot completely exclude that the drop in activity is due to a lower abundance of the mutant receptor. To further characterise the impact of the L72A mutation, the variant was purified and we conducted a differential HDX-MS experiment using the tβ1AR L72A mutant, in the presence of the agonist isoprenaline and the antagonist cyanopindolol. Interestingly, our data for the tβ1AR L72A mutant demonstrated no observable difference in deuterium uptake between the bound and unbound state of the mutant on ICL1, irrespective of whether an agonist or antagonist was present (Fig. 5H, I). However, the relative deuterium uptake for unbound state of tβ1AR L72A mutant is significantly higher compared to that of the unbound of tβ1AR. This observation indicates that L72A mutation disrupts the connection between H8 and ICL1 by destabilising the ICL1 itself (Supplementary Figs. 17–22). The interaction between ICL1 and H8 in class A GPCRs has been previously identified in a study investigating GPCR activation mechanisms. In the same research, the Leucine residue on ICL1 was recognised as a key player in establishing contact with H8[41].

## Importance of β1AR intracellular loop 1 (ICL1) in allosteric activation of miniGs

Having determined the influence of ICL1 on the conformational dynamics of tβ1AR, we carried out differential HDX-MS experiments by coupling tβ1AR to miniGs and subsequently binding it to an agonist, antagonist, and partial agonist. Our data suggest that all ligands induce a similar conformational pattern of the receptor when complexed with miniGs (Supplementary Fig. 15). Notably, we observed a protection on the ECL side, which encompasses ECL1 and the apex of TM1 as well as ECL2 and TM5 regions identified as ligand binding sites. The results also indicate a decrease in deuterium uptake on the cytoplasmic side of the receptor ICL2 and TM5/ICL3/TM6 in response to miniGs binding. These results corroborate previous structural studies on GPCR/miniGs complexes, thereby representing the primary contacts formed with miniGs[42]. We further observed protection on ICL1 and H8, which we hypothesise contributes to the stabilisation of the tβ1AR/miniGs complex by locking the receptor in a stable conformation. Nevertheless, the potential involvement of ICL1 and H8 in the coupling interface cannot be excluded (Fig. 6A). The full agonist elicited larger differences in deuterium uptake in ICL2 and TM5/ICL3/TM6 in comparison to the partial agonist and antagonist (Supplementary Fig. 21). In addition, a significantly higher number of peptides displayed an effect in the isoprenaline-bound state, implying an enhanced stabilisation of the tβ1AR/miniGs complex upon agonist binding.

We also looked at the dynamics induced in miniGs upon coupling with tβ1AR. Interestingly, we noted a protection on helix 5 (H5), S2-S3, and H2 across all three complexes, which validates the central role of H5 as the interface contact with β1AR. Intriguingly, isoprenaline stimulated increased deuterium uptake on H1, P-loop, and S5-HG, a phenomenon absent in the presence of the antagonist. The increased dynamics observed in the H1, and P-loop region can be attributed to a broken connection with H5 upon β1AR binding[43]. Furthermore, the observed deprotection on S5-HG and H1 is likely indicative of a disrupted contact between S5-HG, H1, and the nucleotide-binding site, culminating in GDP release[43].

To investigate whether or not the L72A mutation impacts on receptor/miniGs coupling, we performed differential HDX-MS experiment with the isoprenaline bound tβ1AR L72A mutant in the presence and absence of miniGs (Fig. 6B). Our results demonstrated that the L72A mutant could still bind to miniGs, inducing the same conformational changes on the ICL and the ECL side as tβ1AR with the exception for ICL1 and H8. Previously observed protection on ICL1 and H8 for the tβ1AR in complex with miniGs, has been abolished by the presence of the L72A mutation. Relative deuterium uptake for ICL1 is significantly higher for the tβ1AR L72A mutant than for tβ1AR (-7 Da and -3 Da, respectively), indicating increased dynamics of the loop in L72A mutant. Additionally, the mutation of L72A has abolished the increased

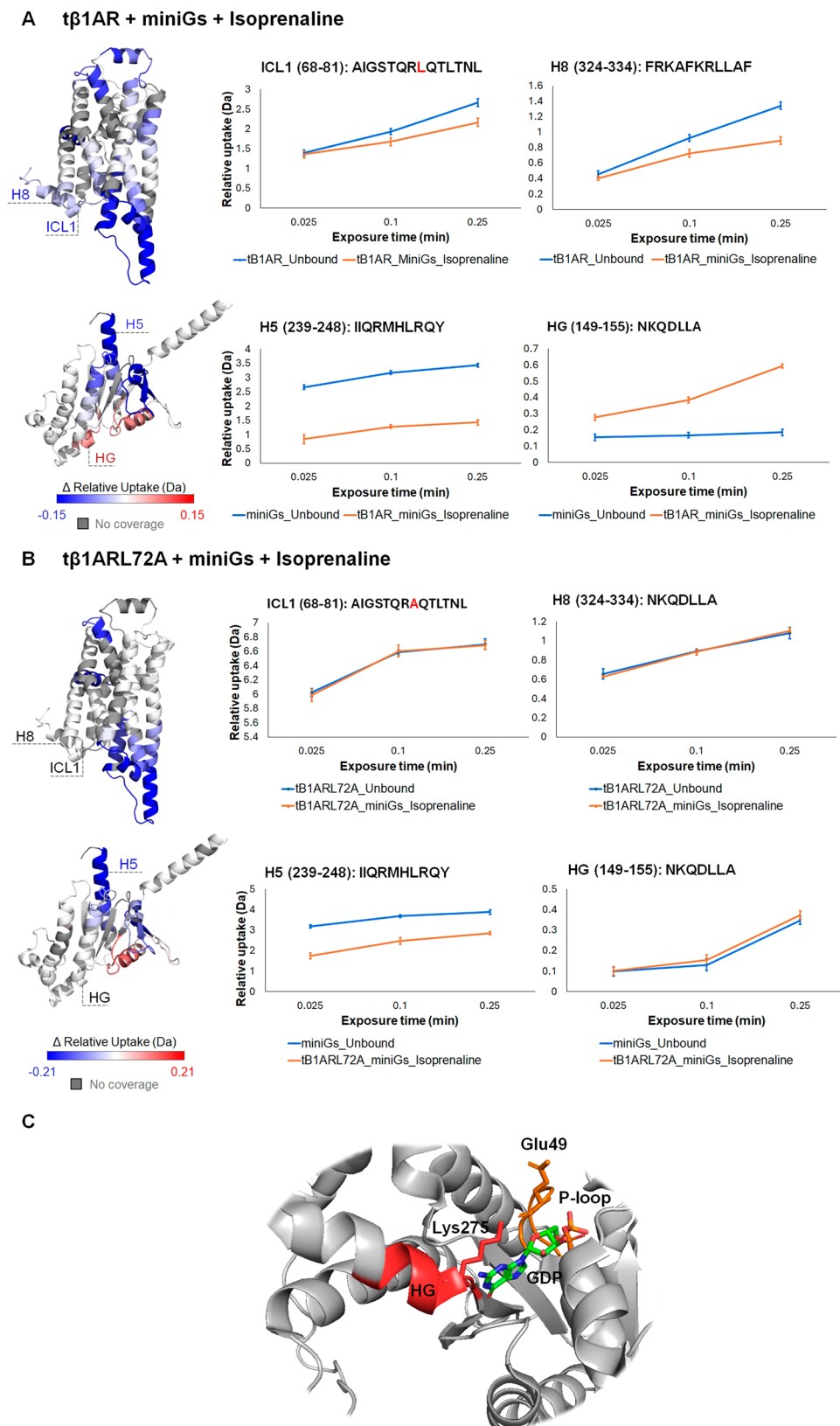

dynamics of the S5-HG region of miniGs which was observed with the tβ1AR. The level of protection observed on H5 of miniGs, which represents the primary binding epitope to the β1 adrenergic receptor (β1AR), is significantly reduced in the mutated L72A receptor. Our hypothesis suggests that the alterations observed in ICL1 and H8 of the tβ1AR L72A mutant could impede the receptor's ability to adopt a stable conformation necessary for interacting with H5 of miniGs. However, we cannot discount the possibility that effective coupling to the G protein may involve direct engagement with ICL1 and H8. As a result, there is a decreased difference in deuterium uptake on H5 observed in the coupling experiment with the L72A mutant (Fig. 6B) The interaction between H8 of class A GPCRs and H5 of miniGs[44], could

**Fig. 6 | ΔHDX of tβ1AR/miniGs/isoprenaline and tβ1AR L72A/miniGs/iso-prenaline complex. A** Regions showing significant difference in HDX for tβ1AR/miniGs complexes with agonist isoprenaline with uptake plots displaying the effect on ICL1 and H8 of tβ1AR and on H5 and HG of miniGs. **B** Regions showing significant difference in HDX for tβ1AR L72A/miniGs complex with isoprenaline with Uptake plots displaying the effect on ICL1 and H8 of tβ1AR and on H5 and HG of miniGs. Results from both experiments are mapped onto β1AR structure (modelled by the use of PDB structures; 2VT4 chain A and 6IBL chain A) and miniGs crystal structure.

**C** The view of miniGs with bound GDP (PDB: 6EG8)[61] and with highlighted residues (Lys275, Glu49) that are part of salt bridge between HG and P-loop. The red colour indicates deprotection of HG. Relative deuterium uptake shows cumulative data across all time points for every receptor-ligand complex. Data are presented as mean values ± SD, where error bars represent the standard deviation at each time point. Each measurement is based on three technical replicates (*n* = 3). Source data are provided as a Source Data file.

be affected in our experiments with the tβ1AR L72A mutant. Furthermore, a change in this interaction could affect contact re-organisation within miniGs, that are essential for its activation. Lack of deprotection on S5-HG indicate that contact between H5 and S5-HG was not disrupted in the L72A complex as it is for tβ1AR. In the inactive state, residues located on HG of miniGs contact GDP, whereas in the active state (GPCR bound) this contact is disrupted to promote GDP release[43]. Studies performed on β2AR report that conformational changes on H5 of the Gα subunit affect the nucleotide binding region by directly impacting on S5 and HG helix, which interacts with guanine ring of GDP via two residues (Lys275 and Asp277)[45]. Increased disorder of those residues as well as increased mobility of HG were observed in GPCR-mediated activation[45]. Additionally, one of these residues (Lys275) located on HG is a part of important salt bridge formed between HG and P-loop (Glu49) (Fig. 6C), and whose disruption is a key element facilitating the GDP release[46]. Due to lack of deprotection on HG, our data demonstrate that contact between HG and GDP was not compromised in our experiment with tβ1AR L72A mutant indicating that the L72 residue on ICL1 of tβ1AR has an impact on stability of the GDP-bound state by directly affecting the dynamics of HG helix of miniGs.

## Discussion

Here, we have demonstrated the unique utility of our HDX-MS approach in defining conformational fingerprints of tβ1AR upon different ligands binding. We showed that such fingerprints can be used to reveal the allosteric modulation of receptor dynamics in response to external stimuli. We have exemplified this method by monitoring the change in dynamics induced by a range of compounds, such as full/partial agonists and antagonists. Our results highlight that compounds with similar pharmacological function impose, or stabilise, similar conformational signatures across the receptor, thereby allowing the mode of action of compounds to be discerned in a single experiment. This provides highly valuable insights for downstream pharmaceutical research, allowing early hits which have the desired pharmacology to be progressed and allows more informed decisions as compounds undergo further SAR development towards candidate drugs. This is particularly important for drugs which are being developed to invoke a biased receptor response, whereby the development of any lead compound must maintain a strict propensity towards the desired signalling pathway. This gives further insights into structural aspects which are still widely debated, such as partial agonism. To exemplify this, biased ligands such as isoprenaline displayed reproducible deprotection on ICL3, whereas norepinephrine did not, suggesting that ICL3 deprotection could be used as a marker to confirm biased activity.

The findings of this research underscore the crucial role of ICL1 in receptor activation. Previous mutagenic analysis of residues in ICL1 of class B1 GPCRs has demonstrated that mutations in this loop region can significantly modulate receptor activation[47,48]. Concurrent molecular dynamic simulations of the Calcitonin Receptor-Like Receptor revealed minor movement of ICL1, succeeded by rearrangements of ICL4 (the junction between TM7 and H8). Certain residues within ICL1 orientate towards H8, while residues of H8 are centrally directed into ICL1, suggesting a potential interaction between ICL1 and H8. These alterations are viewed as initial steps of receptor activation, occurring

prior to movement of TM5 and TM6[47]. Further research on the active and inactive state structures of GPCRs from each class has reported that state-specific contacts are established not only to transmembrane helices but also to H8 and ICL1. Consequently, the active and inactive states of class A GPCRs are stabilised by rerouting contacts between residues located on transmembrane helices 1–7 as well as ICL1 and H8, with one of the residues identified as a state switch[41,49].

In our studies of tβ1AR dynamics, we observed repeated destabilisation of ICL1 following binding of agonists. MD simulations, in combination with characterising residue conservation across several homologues for this loop, identified residue L72 as forming key interactions with conserved hydrophobic residues of H8. Mutational analysis in cell-based assays further support a potential functional importance of ICL1, essentially causing an almost complete loss of signalling in the single-point variant. We cannot rule out that lower expression levels of the L72A may contribute to the magnitude of this observation. Nonetheless, HDX analysis using the L72A receptor variant revealed that conformational differences previously observed for tβ1AR ICL1 between agonist bound/unbound receptor were completely abolished. This suggests that mutation of this residue causes increased dynamics of the loop that results in contact disruption between ICL1 and H8. Studying the addition of miniGs into HDX experiments, clear protection at expected residues on the receptor confirmed that this variant is still able to form interactions with the G protein. The protection previously observed for the tβ1AR on ICL1 and H8 has been compromised in the L72A mutant of tβ1AR. This suggests that the increased dynamics of ICL1 caused by the mutation disrupt the formation of contacts between the receptor and H5 of miniGs. As a result, the internal reorganisation of contacts within miniGs, particularly between H5 and other areas of miniGs, may be significantly disrupted. Supporting this theory is the experimental observation that the tβ1AR L72A mutant does not show deprotection on the HG of miniGs that directly interacts with nucleotide binding site. We believe our data provides evidence that mutation of residue L72 on ICL1 of tβ1AR impacts GDP release and ultimately miniGs allosteric activation. An interesting extension to this work would be to study the impact of these altered dynamics in the mutated receptor on β-arrestin coupling, which may provide mechanistic insight into biased signalling for this receptor. Overall, our data provide insights into GPCR–G protein activation as well as GDP release mechanism.

## Methods

### Expression and purification of tβ1AR and tβ1AR_L72A mutant

Meleagris gallopavo tβ1AR construct (β114-E130W) harbours various thermostabilised point mutations and truncations at the N terminus, inner loop 3 and C terminus (β44-m23) as previously described[19] and an additional set of mutations to aid the purification step and retain functionality. The E130W mutation described here facilitated purification of the receptor in its apo state by stabilising the intrahelical interactions between transmembrane helices TM3, TM4 and TM5. The constructs (tβ1AR and L72A mutant isoform) were over expressed in insect cells (sf9, Invitrogen, 11496015) using the Bac-to-Bac baculovirus expression system (Thermo Fisher). Expression vector pfastBac1 (Thermo Fisher) was used to generate tβ1AR and mutant isoform (L72A) tβ1AR at a viral titre of multiplicity of infection of 0.5.

Cell pellets were thawed on ice, then resuspended in ~25 mL of buffer (20 mM Tris−HCl (pH8), 1 mM EDTA, protease inhibitor Roche); 10 mL/g of dry pellet weight. The buffer/cell mixture was repeatedly aspirated using a pipette until the homogenous suspension was achieved. The suspension was added to the cell disruptor (microfluidics; Quadro engineering) for lysis. The lysate was passed through the microfluidizer twice. Lysed cells were then collected from the cell disruptor and distributed between 40 mL tubes suitable for ultracentrifugation (Avanti with pre-chilled JA25.5 rotor). The lysate was spined at $9000 \times g$ for 20 min at 4 °C. The supernatant containing clarified membrane suspension was collected in 250 mL pot. Then the clarified cell lysate was transferred to the tubes and centrifuged at 170,000 × g for 2 h at 4 °C (Avanti Optima XPN−80). After the spin, the brown pellet was visible in the tubes and the supernatant was discarded. Following that, pellets were transferred to the pre-chilled homogeniser and 40 mL resuspension buffer was added. The pellets were passed through the shearing process for ~20 times. After that, homogenised samples were transferred to the ultracentrifuge tubes and spined at $170,000 \times g$ for 1 h at 4 °C. The supernatant was discarded, and 40 mL of wash buffer was added to the brown pellet. The samples were homogenised as described before. Resuspended pellet was once more ultracentrifuged. Following that, the brown pellet was resuspended in the final resuspension buffer to a protein concentration of 10 mg/mL.

The cell membranes were solubilized in 20 mM Tris−HCl (pH 8), 350 mM NaCl, 3 mM imidazole and 1.5% w/v n-dodecyl-β-d-maltopyranoside (DDM; Anatrace). The supernatant was obtained following ultracentrifugation at 175,000 × g for 1 h and loaded onto a HiTrap TALON crude column (GE Healthcare) for affinity enrichment, pre-equilibrated in wash buffer 20 mM Tris−HCl (pH 8), 350 mM NaCl, 3 mM imidazole and 0.05% DDM. The receptor was eluted in using buffer composed of 20 mM Tris−HCl (pH 8), 350 mM NaCl, 250 mM imidazole and 0.05% DDM. The protein was quantified on a SDS PAGE. The fractions containing receptor were pooled and concentrated to a final concentration of 2−3 mg/mL using an Amicon centrifugal filter with a molecular weight cut-off of 50 kDa.

### Expression and purification of miniGs

The engineered minimal G protein, miniGs construct E392A was expressed in *E.coli* cells. The pellet from *E.coli* culture was resuspended in buffer A (40 mM HEPES, pH 7.5, 100 mM NaCl, 10 mM imidazole, 10% v/v glycerol, 5 mM MgCl₂, 50 µM GDP) with protease inhibitors, DNase I, lysozyme and DTT. The cells were lysed in the microfluidizer followed by lysate centrifugation at 38,000 × g for 45 min. Collected supernatant was filtered and loaded onto TALON column. The column was washed with buffer B (20 mM HEPES, pH 7.5, 500 mM NaCl, 40 mM imidazole, 10% v/v glycerol, 1 mM MgCl₂, 50 µM GDP). The protein was eluted with buffer C (20 mM HEPES pH 7.5, 100 mM NaCl, 500 mM imidazole, 10% v/v glycerol, 1 mM MgCl₂, 50 µM GDP). Fractions containing the protein were concentrated and subjected to desalt with HiTrap desalting column in buffer D (20 mM HEPES, pH 7.5, 100 mM NaCl, 10% v/v glycerol, 1 mM MgCl₂, 10 µM GDP). After desalting, protein concentration was measured and DTT (1 mM), GDP (100 µM) and TEV protease (in-house, 4 mg/mL) to give a TEV:miniGαs ratio of 1:20 w/w, were added to the sample in order to cleave off the histidine tag. Sample was incubated on ice for 2 h, which was followed by reverse IMAC purification with TALON (cobalt) column equilibrated with buffer D. Protein was concentrated to the final concentration of 0.4 mg/mL.

### Ligand solution preparation

Ligands were purchased from Combi blocks (carazolol), Enzo life science (cyanopindolol), Sigma-Aldrich (salbutamol and dobutamine), BLD Pharmatech (isoprenaline and orciprenaline) and Carbosynth (norepinephrine). Isoprenaline and orciprenaline were dissolved in water, whereas rest of the ligands were dissolved in 100% DMSO until the desired concentration was achieved. Concentration of DMSO was kept below 10% for all ligand-protein mixtures.

### Deuterium labelling of β1AR/ Ligand complexes

Purified β1AR at the concentration of 11 µM was used for all HDX experiments. The equilibration buffer (E) was composed of 20 mM Tris−HCl, pH 8, 0.35 M NaCl, 3 mM imidazole, 0.05% DDM. The quench buffer (Q) was composed of 50 mM K₂HPO₄, 50 mM KH₂PO₄, 0.1% DDM, 100 mM TCEP. The labelling buffer (L) was the same composition as (E) buffer but was prepared using D₂O (99.8%). The protein/ligand ratio was calculated based on (Eq. 1) in order to ensure 99% binding occupancy of the receptor (Supplementary Table 2)[50]. Deuterium labelling was performed by diluting 3.5 µL of protein (with or without ligand) in 32 µL of labelling buffer. The protein sample was incubated for various time points (15 s, 2 min, 30 min and 120 min at room temperature) and then quenched with 35 µL of buffer Q. Sample preparation (labelling and quenching) and injection was performed by an online robotic autosampler (LEAP HDX automation manager).

$$[RL] = \frac{K_d + R_0 + L_0 - \sqrt{(K_d + R_0 + L_0)^2 - 4R_0L_0}}{2[R_0]} \tag{1}$$

Parameters in the equation are as follow: $K_d$ – dissociation constant, $[R_o]$ – concentration of the receptor before labelling, $[L_o]$ – concentration of the ligand before labelling, $[RL]$ – fractional occupancy.

### Deuterium labelling of tβ1AR/miniGs/Ligand complexes

The coupling experiment was performed with tβ1AR and miniGs at a molar ratio of 1:1.2 (11.5 µM: 13.8 µM), respectively. Sequence coverage obtained for miniGs was 92.4% with a redundancy of 7.26. The equilibration buffer (E) was composed of 20 mM HEPES, pH 7.5, 100 mM NaCl, 10% v/v glycerol, 1 mM MgCl₂, 10 µM GDP. The quench buffer (Q) was composed of 50 mM K₂HPO₄, 50 mM KH₂PO₄, 0.1% DDM, 100 mM TCEP. The labelling buffer (L) was the same composition as (E) but prepared using D₂O (99.8%) and 0.05% DDM was added. The protein/ligand ratio used was calculated based on (Eq. 1) in order to ensure 99% binding occupancy of the receptor (Supplementary Table 2)[50] Deuterium labelling was performed by diluting 3.5 µL of protein (with or without ligand) in 32 µL of labelling buffer. The protein sample was incubated for various time points (15 s on ice, 60 s on ice and 15 s in room temperature) and then quenched with 35 µL of buffer Q. Deuterium labelling and quenching of samples was performed manually on ice which was followed by manual injections.

### LC−MS workflow parameters for all HDX samples

Injected onto a nanoACQUITY UPLC system samples were digested using a dual protease type XIII/pepsin column from NovaBioassays. HPLC run time was 11 min using a flow rate of 40 µL/min with gradient that started with 8% B increased to 55% B within 8 min and then quickly increased to 85% B in 0.5 min, held for 1.5 min and decreased back to 8%. The C18 trap column (ACQUITY UPLC®BEH 1.7 µm, Waters) and a C18 column (ACQUITY UPLC®BEH, 1,7 µm, 1.0 × 100 mm, Waters) were utilised for the experiments. For HPLC analysis, buffer A was composed of H₂O with 0.1% formic acid, buffer B was acetonitrile with 0.1% formic acid at flow rate of 40 µL/min. The gradient started with 8% B increased to 55% B within 8 min and then quickly increased to 85% B in 0.5 min, held for 1.5 min and decreased back to 8%. Analysis was performed by applying normal collision energy (CE) and look up table CE, which is a time-dependent CE gradient. In this method, the optimal CE is calculated for a set of peptides based on reference samples which has been shown to provide increased sequence coverage and redundancy for HDX experiments[51]. The mass range for MS was *m/z* 100−2000 in positive ion mode on the Synapt G2-Si mass spectrometer with ESI source and ion mobility cell, coupled to ACQUITY UPLC with HDX

**Table 1 | Overview of system setup**

| System | Simulation box dimensions (Å³) | #atoms | #water molecules | [NaCl] (mM) | #lipids | Lipid type |
|---|---|---|---|---|---|---|
| tβ1AR apo | 80 × 80 × 100 | 61230 | 12708 | 150 | 137 | POPC |
| L72A apo | 80 × 80 × 100 | 61200 | 12701 | 150 | 137 | POPC |

Automation technology (Waters Corporation, Manchester, UK). Leucine Enkephalin was applied for mass accuracy correction and sodium iodide was used as a $m/z$ calibrant for the mass spectrometer. The trap and analytical column were kept at 0 °C inside the HDX manager. Between every injection, clean blank was injected to remove any carryover. The data for each time point were obtained in three replicates. Data processing and analysis was performed processed using MassLynx (Waters), (ProteinLynx Global Server 3.0.2 © Waters, 2013) was used to analyse the MS data of unlabelled peptide and generate peptide libraries for each target protein. DynamX 3.0 (Waters) was used to analyse and quantify the deuteration for each peptide and Deuteros[34] was used to assess statistically significant differences in deuterium uptake for peptides in two different conditions. It is noted that the residue numbering on the HDX plots refers to the sequence of tβ1AR used in the lab which differs from the WT sequence.

### Multiple sequence alignment and sequence logos
The β1AR wild *turkey* sequence available on uniprot[52] was used as reference sequence (uniprot ID P07700). A Blast search was performed through the uniprot website and the top 1000 hits were used as input for a multiple sequence alignment using Clustal Omega through EBI[53]. The resulting multiple sequence alignment was used for creating the sequence logos through WebLogo3[54].

### MD simulations
Chain A from the 4AMJ[40] pdb file of the *turkey* β1AR was used as a model for the tβ1AR apo state (resolution 2.30 Å). Both protein and water ascribed to chain B were deleted along with all other small molecules bound, apart from the sodium ion and the water molecules bound to chain A. The protein was prepared for simulation using the standard protein preparation workflow in Maestro[55]. The prepared protein structure was aligned to the 2y02 structure from the Orientations of Proteins in Membranes database[56] to get a reasonable initial membrane position. An 80 Å × 80 Å pure POPC (1-palmitoyl-2-oleoyl-glycero-3-phosphocholine) bilayer was constructed around the protein using the Desmond[57,58] System Builder and the system was solvated with SPC water molecules, neutralised and further ions were added to a concentration of 150 mM NaCl. The OPLS4 force field[59] available in Maestro/Schrodinger was applied. System setup details are provided in Table 1.

The single-point L72A mutation was introduced using the Mutate function in Maestro and the simulation system was constructed as described above for the tβ1AR system.

The systems were equilibrated in six steps, slowly increasing the temperature and reducing the force constants on the different components, generally following the default relaxation protocol for membrane protein systems in Desmond. Production runs likewise followed the default setup for Desmond simulations, with a temperature of 310 K and standard atmospheric pressure in the NPγT ensemble (constant particle number (N), pressure (P), lateral surface tension (γ), and temperature (T)). For each system 1 × 200 ns and 2 × 500 ns production runs were performed, starting from different seeds.

Analysis was performed using in-house Tcl scripts through VMD[60] and structural images in panels 3B-3E were created using VMD. The plot in panel 3 F was created using seaborn.

### cAMP accumulation assay
CHO cells from Merck (85051005) were maintained in DMEM/F12 cell culture medium supplemented with 10% fetal bovine serum and 1% L-glutamine, at 5% $CO_2$, 37 °C. Cells were grown to 70–80% confluency before transfection with β1AR constructs using FuGENE HD reagent (Promega) according to manufacturer's instructions. The following day, transfected CHO cells were suspended in assay buffer (Hank's balanced salt solution containing 5 mM HEPES, 0.1% w/v BSA and 500 mM 3-isobutyl-1-methylxanthine, pH 7.4) before incubation with increasing concentrations of noradrenaline or isoproprenaline for 1 h at room temperature. After 1 h, cAMP levels were measured using the cAMP Gs HiRange HTRF kit (Perkin Elmer) according to manufacturer's instructions. Fluorescence resonance transfer (FRET) was detected on a PHERAstar plate reader (BMG) and FRET ratios used to determine cAMP levels from a cAMP standard curve from the same experiment. EC50 and Emax were estimated from concentration-response curves using a variable four-parameter logistic equation in Prism (v8; GraphPad).

### Statistical analysis of HDX data
The differential hydrogen-deuterium exchange mass spectrometry (HDX-MS) data sets obtained in this research were analysed statistically to assess significant structural and dynamic changes in specific regions or peptides of the protein. These changes are characterised by differences in centroid m/z values or deuterium incorporation rates during the HDX experiment. The statistical analysis of the data sets was performed using the Deuteros 2.0 software developed by Lau et al.[34]. The hybrid significance test with 99.0% confidence interval was employed to all data sets which are presented in Woods plots.

### Reporting summary
Further information on research design is available in the Nature Portfolio Reporting Summary linked to this article.

## Data availability
The HDX MS raw data files generated in this study have been deposited in the ProteomeXchange via the PRIDE database under accession code PXD051297. The HDX summary tables and all deuterium uptake plots complying to the community-based recommendations are provided in the Supplementary Information Supplementary Tables 4–10 and Supplementary Figs. 17–22, respectively. The MD data generated in this study have been deposited in the Figshare database under 26125750. Source data are provided with this paper.

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

## Acknowledgements

A.P. was supported by an EPSRC Fellowship (EP/V011715/1) and a BBSRC grant (BB/V006487/2). This work was supported by EPSRC CASE Studentship co-funded by OMass Therapeutics to A.P. and J.T.

## Author contributions

J.T. and P.K. purified β1AR. J.T. purified miniGs J.T. and A.P. designed the HDX-MS experiments, with input from J.T.S.H., H-Y.Y. and J.T. carried out all HDX-MS experiments and data analysis. M.M. performed multiple sequence alignment, sequence logos and MD simulations. K.G. and K.S. performed cAMP accumulation assay. F.Q. and M.S. performed characterisation experiments on transfected cells. K.H. helped in optimising HDX-MS method. A.J., J.T.S.H. and A.P. supervised the project. J.T., J.T.S.H. and A.P. wrote the paper with contributions from all authors.

## Competing interests

P.K., M.M., K.S., K.G., F.Q., M.S., H-Y.Y., A.J. and J.T.S.H. are shareholders of OMass Therapeutics. The authors declare no competing interests.
