## [Peer Review File · Nature Communications]

Ligand-induced conformational changes in the β 1-Adrenergic Receptor Revealed by Hydrogen-Deuterium Exchange Mass SpectrometryREVIEWER COMMENTS

Reviewer #1 (Remarks to the Author):

The manuscript by Toporowska et al., probes the conformational dynamics of activation and G-protein-coupling of the Beta1AR using HDX-MS and MD simulation. Using a set of pharmacological different ligands, this study draws interesting insights into receptor activation, especially a significant role of ICL1. Overall, the study is very well designed and represents an important advance in the field of GPCR dynamics and signaling with very well-designed experiments and data presentation. I strongly recommend publication of this manuscript, and I have only a couple of minor comments that the reviewer may consider during revision.

1. The cAMP data on L72A mutant is truly intriguing and I wonder if it is also compromised in terms of the transducer i.e. beta-arrestin? It should be a straightforward experiment and may provide either distinct contribution of this residue in selective coupling of G-protein with implication for signaling-bias or may underscore conserved role in transducer-coupling making it interesting either way. I also did not find surface expression of this mutant in the main file or in supplemental, but I assume that the authors have measured it, and it is expressed at comparable levels as the wild-type. It would be important to mention this somewhere in the text.

2. It would be important to deposit the MD trajectories in one of the databases such as GPCR-MD which is becoming a common practice now, and mention this somewhere in the text.

Arun K. Shukla, PhD

Reviewer #2 (Remarks to the Author):

In this manuscript, the authors describe the use of HDX and other allied methods (mutagenesis, simulations) to study binding of the GPCR A2AR to a variety of ligands with different functions. This is an elegant study that is well executed and the manuscript describes the experiments and their implications for GPCR function well. The authors should be congratulated for including details here regarding the optimisation process for HDX of this GPCR – as this will be a useful resource to enable this methodology to be widely deployed. This raises the impact of the work significantly, as not only are there novel biological insights presented here, but there are also significant methodological advances that will enable others to take advantage of this technology. I recommend publication of the manuscript after the following very minor comments are addressed, which mainly relate to data presentation and editorial comments.

- Page 4, final introduction paragraph. What do the authors mean by ‘opposing dynamics’?

- In Fig 1A-C, can the authors indicate the helices? There is discussion in around specific TMs and Ser residues that are protected and it is unclear which regions are of importance. Further structural features such as the 'lid' may also be indicated here.
- In FigA-C are the relative uptake colours shown for a specific timepoint or cumulative data over all timepoints? This needs clarification.
- Figure legend for Fig 1 missing mention of panel C
- Page 8 – Why do the authors ascribe differences in uptake between ligands to their different affinities? The data are all acquired under comparable % bound conditions so it is unclear why such differences would be observed. Is this related to relative on/off rates?
- In Supplementary Fig 7 can the authors indicate locations of helices etc? Clearly recognising these regions is key for some of the conclusions the authors draw.
- In Fig 5L there seems to be only one deprotected peptide shown at 0.25 min in the Wood's plot, but there are many regions of deprotection in the structure. Is this because the structure is the summed data over all timepoints? This is unclear and needs commenting on.

Reviewer #3 (Remarks to the Author):

In the manuscript, HDX-MS is used to investigate the dynamic changes of structural segments of the turkey β 1-adrenergic receptor (t β 1AR) after binding of different modulators. In addition to the expected stabilization of the extracellular region around the orthosteric ligand binding pocket, the authors observe different effects on the mobility of intracellular loop 1 (ICL1) depending on the type of compound. In particular, the binding of an agonist seems to induce a higher mobility of ICL1. The authors use molecular dynamics simulations and mutational analyses in combination with functional analyses to study the local dynamics of ICL1 and investigate downstream signaling by measuring the HDX of bound miniGs.

The work sheds light on long-range allosteric effects, which are an important basis for clarifying how different modulators tune the receptor for coupling to G proteins or arrestin. While I find the analysis compelling, I have problems with some of the interpretations of the data. At many points, the manuscript in its current form neglects important insights from the related β 2-adrenergic receptor system that would be helpful in interpreting the ICL1 data, especially in light of very recent work investigating the mechanisms of G protein association or dissociation with the receptor. In addition, the author should clarify the limitations of using miniGs as a tool to study G-protein coupling.

I have listed my remarks below.

Remarks

- Abstract: "Multiple sequence alignments and molecular dynamics simulations indicate that L72 in ICL1 plays an important structural role." This statement seems a bit unclear to me. I guess the former indicates conservation, while the latter dynamics. Please rephrase.
- Introduction (1. Paragraph): "SAR development". Please introduce abbreviation.
- HDX fingerprint of $\text{t}\beta\text{1AR}$ bound to antagonist: There are a number of published papers on the effects of the so-called "lid" on affinity (Sunahara lab) and binding for the β2 -adrenergic receptor (Dror lab) that should be referenced and discussed here.
- "Previous studies have reported that coupling of a GPCR to G protein takes place through the cytoplasmic loops with ICL3 being of particular interest as it is responsible for determining specificity between different G proteins (ref 30)." To my knowledge, this statement does not correctly summarize the observations in ref. 30. If ICL3 were responsible for determining specificity, replacing the ICL3 loops would be sufficient to change the coupling specificity. As the title suggests, the paper rather describes autoregulation by ICL3. With regard to coupling specificity, it states that the "ICL3 tunes signaling specificity by inhibiting receptor coupling to G protein subtypes that weakly couple to the receptor".
- "In contrast to ICL3, enhanced dynamics of ICL2 are likely related to the strength of G protein coupling rather than specificity (ref 5)." I would disagree with this statement given the published work showing different conformations of ICL2 for the β2 -adrenergic receptor coupled to G_s or G_i by NMR (Ma et al, PNAS, 2020, Kobilka lab).
- "GPCRs couple to G protein at the cytoplasmic side mostly involving the end of TM3/ICL2, TM6 and TM5/ICL331." Probably wrong reference. The authors are probably referring to findings from nucleotide-free structures...
- "...as well as the crucial role of ICL3 in defining specificity for distinct G proteins and facilitating biased signaling pathways." The latter statement (signaling bias) should either be explained in more detail or rejected.
- "to assess the ICL1-H8 association". Would "packing" be a more appropriate formulation compared to "association" which refers to a process.
- "showing great fluctuations in the distance (Fig. 5F)." Fig. 5F shows the distributions of distances, indicating a broader distribution for the L72A mutant. It does not show fluctuations.
- "Interestingly, our data for the $\text{t}\beta\text{1AR}$ L72A mutant demonstrated no observable difference in deuterium uptake between the bound and unbound state of the mutant on ICL1, irrespective of whether an agonist or antagonist was present (Fig. 5H, I)." Does this observation indicate that the mutant disrupts the connection to ICL1, but does not destabilize it?
- Fig. 5 C-E. The illustrations could be improved by placing the labels with thin arrows next to the cartoon and perhaps using thinner helices to focus more on the structural features that these illustrations want to emphasize.

- We further observed protection on ICL1 and H8, which we hypothesize contributes to the stabilization of the t β 1AR/miniGs complex by “locking” the receptor in a stable conformation.” Are there also alternative interpretations of the data? Can the authors completely rule out that the ICL1 and H8 have a role during MiniG binding by being part of the interface?

- “We hypothesize that the compromised protection on ICL1 and H8 in the t β 1AR L72A mutant may be indicative of structural rearrangements that prevent the receptor from adopting a stable conformation that allows for interactions with H5 of miniGs.” See last comment. Can the authors rule out that productive coupling to the G protein requires direct contact with ICL1 and H8?

- It would be convenient for the reader to introduce the nomenclature used for “LysX and AspZ”.

Ligand-induced conformational changes in the β 1-Adrenergic Receptor Revealed by Hydrogen-Deuterium Exchange Mass Spectrometry

We would like to express our gratitude to the reviewers for their valuable feedback on our manuscript titled "Ligand-induced conformational changes in the β 1-Adrenergic Receptor Revealed by Hydrogen-Deuterium Exchange Mass Spectrometry". We appreciate the time and effort they dedicated to providing thoughtful comments and suggestions for improving our work. Below, we address each of the reviewers' comments in detail:

Reviewer 1:

The manuscript by Toporowska et al., probes the conformational dynamics of activation and G-protein-coupling of the Beta1AR using HDX-MS and MD simulation. Using a set of pharmacological different ligands, this study draws interesting insights into receptor activation, especially a significant role of ICL1. Overall, the study is very well designed and represents an important advance in the field of GPCR dynamics and signaling with very well-designed experiments and data presentation. I strongly recommend publication of this manuscript, and I have only a couple of minor comments that the reviewer may consider during revision.

1. The cAMP data on L72A mutant is truly intriguing and I wonder if it is also compromised in terms of the transducer i.e. beta-arrestin? It should be a straightforward experiment and may provide either distinct contribution of this residue in selective coupling of G-protein with implication for signaling-bias or may underscore conserved role in transducer-coupling making it interesting either way.

We thank the reviewer for this suggestion, and we agree that it would be a fascinating extension of the work to also consider any potential role in biased signalling, although we feel that this investigation falls outside the scope of this work. For these experiments we would want to couple any pharmacological data to dynamics observed via HDX to fit with the current theme of the paper. We agree that the pharmacology experiments would be relatively straightforward in theory, however we do not have this capability currently established in-house and would therefore take a considerable amount of assay development work. Furthermore, the structural side of this proposed work would ideally use a phosphorylated receptor for comparison, which is required prior to beta-arrestin engagement and signalling. Extensive protein biochemistry and subsequent HDX work would be required which we feel would be better suited to a separate study. Nonetheless, we certainly feel that this is a great suggestion and would be a worthy extension to the work in the future and have therefore added this to the final paragraph of the conclusions to encourage this as a focus of follow-up investigations.

I also did not find surface expression of this mutant in the main file or in supplemental, but I assume that the authors have measured it, and it is expressed at comparable levels as the wild-type. It would be important to mention this somewhere in the text.

Thank you for highlighting this point, which indeed is an important consideration for these experiments. We have certainly performed these experiments and have included the data in the submitted manuscript – within the supplementary information (Supplementary Figure 5). The data presented allowed us to confirm that the mutant receptor was indeed expressed and correctly folded (following DDM solubilisation), however, the expression levels between L72A and parent receptor do appear different. We understand that these differences may contribute to the activity differences observed and as a result, we have highlighted this point both in the relevant results section, as well as the conclusion to the paper. We copy these statements below:

In the main text where the results are presented, we state:

“In support of our predictions, the L72A variant showed a significantly impaired response to agonist, almost completely abolishing the resulting cAMP levels relative to parent tβ1AR for both tested agonists (Fig. 5G). Receptor expression in membrane preparations of CHO cells transiently transfected with tβ1AR and L72A β1AR was determined using Western blot analysis, which confirmed that the L72A receptor was successfully expressed and not misfolded, but that lower expression levels were observed for the mutant compared to the parent receptor (Supplementary figure 5).”

In the conclusions, we reiterate:

“Mutational analysis in cell-based assays further support a potential functional importance of ICL1, essentially causing an almost complete loss of signalling in the single point variant. We cannot rule out that lower expression levels of the L72A may contribute to the magnitude of this observation”.

2. It would be important to deposit the MD trajectories in one of the databases such as GPCR-MD which is becoming a common practice now, and mention this somewhere in the text.

We thank the reviewer for the reminder to deposit the MD simulation trajectories and make them publicly available. GPCRmd is an excellent resource, however, given our file formats, which in Desmond/Schrodinger differ somewhat from more widely used MD engines, we have had great difficulty making the data available through the database. Thus, we have instead opted to make all the MD simulation data available on FigShare; using Springer Nature’s partnership with the service. In this way, the data should be directly associated with the manuscript and directly downloadable to anyone interested. This has been noted in the “Data Availability” section of the manuscript.

Reviewer 2:

1. Page 4, final introduction paragraph. What do the authors mean by ‘opposing dynamics’?

We thank the reviewer for pointing out this. In the context of our study, 'opposing dynamics' in the intracellular loop 1 (ICL1) of the receptor refer to the observation of contrasting structural effects: deprotection and protection. Deprotection (increased deuterium uptake) was repeatedly observed in agonist-bound state, whereas protection (decreased deuterium uptake) was observed in antagonist-bound state. These opposing dynamics suggest that ICL1 serves as a structurally essential component in regulating receptor function by driving both activation and deactivation processes. We have corrected the text to have "*opposing structural effects (deprotection vs protection)*" which we hope it clarifies this point.

2. In Fig 1A-C, can the authors indicate the helices? There is discussion in around specific TMs and Ser residues that are protected, and it is unclear which regions are of importance. Further structural features such as the 'lid' may also be indicated here.

Figures 1A-C have been revised to include reviewer's suggestion.

3. In FigA-C are the relative uptake colours shown for a specific timepoint or cumulative data over all timepoints? This needs clarification.

In Figures 1A-C, the results are mapped onto the β 1AR structure, displaying the relative deuterium uptake. These figures present cumulative data across all time points for each receptor-ligand complex. Throughout the manuscript, all figures with results mapped onto the receptor structure display cumulative data across all tested time points. This has been now clarified in the figure legend.

4. Figure legend for Fig 1 missing mention of panel C.

The missing in panel C in Figure 1 has now been added to the legend.

5. Page 8 – Why do the authors ascribe differences in uptake between ligands to their different affinities? The data are all acquired under comparable % bound conditions so it is unclear why such differences would be observed. Is this related to relative on/off rates?

The observed differences in deuterium uptake between ligands, despite comparable % bound conditions, suggest that factors beyond occupancy are influencing these effects. One plausible explanation is the relative on/off rates of the ligands. Ligands with different affinities typically have varying on and off rates, which can impact the receptor-ligand complex's dynamic behaviour. Higher affinity ligands may have slower off rates, leading to more stable interactions and significant conformational changes or stabilization of specific receptor states. Conversely, lower affinity ligands may have faster off rates, resulting in less stable interactions and different receptor dynamics. These kinetic differences provide a mechanistic basis for the observed uptake variations, independent of % bound conditions.

The following section has been now included in the text:

“Higher affinity ligands generally stabilise receptor conformations more effectively, leading to reduced hydrogen-deuterium exchange. This stabilisation effect occurs even under comparable % bound conditions because high-affinity ligands tend to have

slower off rates, maintaining their interaction with the receptor for longer periods and inducing greater protection. Thus, the observed differences in deuterium uptake likely reflect the varying abilities of ligands to stabilise the receptor, influenced by their relative on/off rates and binding strengths.”

6. In Supplementary Fig 7 can the authors indicate locations of helices etc? Clearly recognising these regions is key for some of the conclusions the authors draw.

In Supplementary Figure 7 and all other Wood's plots, the significant effects are now labelled with the corresponding regions of the protein.

7. In Fig 5L there seems to be only one deprotected peptide shown at 0.25 min in the Wood's plot, but there are many regions of deprotection in the structure. Is this because the structure is the summed data over all timepoints? This is unclear and needs commenting on.

As per comment 3, the results mapped onto receptor structure display relative deuterium uptake summed over all tested time points. The Woods's plot in the figure shows only 0.25 min time point highlighting the lack of effect on ICL1 region. This has been now clarified in the figure's legend.

Reviewer 3:

1. Abstract: “Multiple sequence alignments and molecular dynamics simulations indicate that L72 in ICL1 plays an important structural role.” This statement seems a bit unclear to me. I guess the former indicates conservation, while the latter dynamics. Please rephrase.

We have now rephrased the abstract to include: “Multiple sequence alignments indicate that L72 in ICL1 is highly conserved across different species, suggesting its evolutionary importance. Additionally, molecular dynamics simulations demonstrate that L72 plays a crucial structural role in maintaining the stability and proper functioning of the receptor.”

2. Introduction (1. Paragraph): “SAR development”. Please introduce abbreviation.

Structure-activity relationship (SAR) has now been included in the manuscript.

3. HDX fingerprint of tβ1AR bound to antagonist: There are a number of published papers on the effects of the so-called "lid" on affinity (Sunahara lab) and binding for the β2-adrenergic receptor (Dror lab) that should be referenced and discussed here.

Two additional references have been added to the manuscript.

*First one: Xu X, Shonberg J, Kaindl J. *et al.* Constrained catecholamines gain β₂AR selectivity through allosteric effects on pocket dynamics. Nat Commun, 2023, 14, 2138.*

Second one: Dror RO, Pan AC, Arlow DH. Pathway and mechanism of drug binding to G-protein-coupled receptors. Proceedings of the National Academy of Sciences of the United States of America. 2011, 108, pp. 13118-13123.

We have included the following section in the text (page 6):

“Recent studies on β 2AR and β 1AR receptors, focusing on the influence of ligand flexibility and the entropic component on catecholamine binding, have unveiled significant insights. These investigations have revealed that the shape and stability of the orthosteric pocket can be influenced by surrounding residues, leading to notable variations in ligand affinity. Specific residues within extracellular loop 2 (ECL2) have been identified as key modulators, capable of reshaping the catecholamine binding pocket and distinguishing between different ligand-bound ensembles⁵⁹. Consistent with this finding, several studies indicate that ECL2 is crucial for ligand specificity and determines the affinity of ligands for the receptor. Research suggests that the binding pathway of different ligands consists of two major steps: the ligand first passes between ECL2 and ECL3, then traverses a narrow passage between ECL2 and TM5, TM6, and TM7 before reaching the main binding pocket⁶⁰. These findings conclude that ECL2 plays key roles in the early stages of ligand recognition and in determining binding kinetics^{5,60}.”

4. “Previous studies have reported that coupling of a GPCR to G protein takes place through the cytoplasmic loops with ICL3 being of particular interest as it is responsible for determining specificity between different G proteins (ref 30).” To my knowledge, this statement does not correctly summarize the observations in ref. 30. If ICL3 were responsible for determining specificity, replacing the ICL3 loops would be sufficient to change the coupling specificity. As the title suggests, the paper rather describes autoregulation by ICL3. With regard to coupling specificity, it states that the “ICL3 tunes signaling specificity by inhibiting receptor coupling to G protein subtypes that weakly couple to the receptor”.

Thank you for bringing up this important point. The paper by Wong et al. (1990) investigates the role of the intracellular loops, particularly ICL3, in G protein coupling specificity. The authors created chimeric receptors combining parts of muscarinic cholinergic receptors and beta-adrenergic receptors to study how these regions influence receptor function. The study showed that replacing the ICL3 of the muscarinic receptor with that of the beta-adrenergic receptor allowed the muscarinic receptor to activate Gs proteins, which it normally does not do. This suggests that ICL3 is a critical determinant in the ability of a receptor to couple with specific G proteins. The study also mentions that ICL3 has a regulatory function, possibly tuning the receptor's signalling by inhibiting interactions with G protein subtypes that do not couple strongly. Therefore, we edited that part in our manuscript suggesting that ICL3 is not solely responsible for determining G protein specificity but rather plays a role in fine-tuning signalling specificity.

5. “In contrast to ICL3, enhanced dynamics of ICL2 are likely related to the strength of G protein coupling rather than specificity (ref 5).” I would disagree with this statement given the published work showing different conformations of ICL2 for the

β 2-adrenergic receptor coupled to Gs or Gi by NMR (Ma et al, PNAS, 2020, Kobilka lab).

The study by Ma et al. from the Kobilka lab, indeed sheds light on the dynamics of Intracellular Loop 2 (ICL2) in the context of G protein coupling. The observation of different conformations of ICL2 for the β 2-adrenergic receptor coupled to Gs or Gi proteins underscores the importance of specificity in influencing these dynamics. This aligns with your argument that specificity plays a significant role. Considering this, we agree that the dynamics of ICL2 are likely influenced by the specificity of G protein coupling rather than strength. We revised the statement in our manuscript to reflect that important point.

We reiterate this here:

“Recent NMR study on β 2AR-Gs and β 2AR-Gi complexes highlighted differences in conformational changes in ICL2 between the two complexes, suggesting that ICL2 may also play a role in G-protein coupling selectivity⁶¹. Additionally, it has been proposed that ICL2 acts as a switch, facilitating G protein binding and interacting with the receptor's conserved DRY motif located on TM3⁵. “

6. “GPCRs couple to G protein at the cytoplasmic side mostly involving the end of TM3/ICL2, TM6 and TM5/ICL331.” Probably wrong reference. The authors are probably referring to findings from nucleotide-free structures.

Thanks for pointing out this. We have changed the reference in the text to include the following:

Hilger D, Masureel M, Kobilka BK. Structure and dynamics of GPCR signaling complexes. Nat Struct Mol Biol. 2018 Jan;25(1):4-12. doi: 10.1038/s41594-017-0011-7.

7. “....as well as the crucial role of ICL3 in defining specificity for distinct G proteins and facilitating biased signalling pathways.” The latter statement (signalling bias) should either be explained in more detail or rejected.

Biased signalling, or functional selectivity, refers to the ability of G protein-coupled receptors (GPCRs) to preferentially activate specific downstream signalling pathways upon ligand binding. This phenomenon arises due to the receptor's ability to interact with multiple intracellular signalling proteins in different ways. Intracellular Loop 3 (ICL3) plays a crucial role in defining specificity for distinct G proteins and can also contribute to biased signalling by modulating the receptor's conformational dynamics to favour certain signalling pathways over others. We have now extended the relevant section in our manuscript to include this additional explanation.

This section now reads:

*“Consistent with these findings, we have observed increased deuterium uptake on ICL3 (**Supplementary Figure 18**) for isoprenaline only, which highlights its ability to*

act as a biased agonist. This underscores the crucial role of ICL3 in defining specificity for distinct G proteins and facilitating biased signaling pathways, where different ligands binding to the same receptor can selectively activate specific downstream signaling cascades, leading to distinct physiological responses³².

8. “to assess the ICL1-H8 association”. Would “packing” be a more appropriate formulation compared to “association” which refers to a process.

The word association has been replaced with packing.

9. “ showing great fluctuations in the distance (Fig. 5F).” Fig. 5F shows the distributions of distances, indicating a broader distribution for the L72A mutant. It does not show fluctuations.

The statement has been edited accordingly to reviewer’s suggestions.

10. “Interestingly, our data for the tβ1AR L72A mutant demonstrated no observable difference in deuterium uptake between the bound and unbound state of the mutant on ICL1, irrespective of whether an agonist or antagonist was present (Fig. 5H, I).” Does this observation indicate that the mutant disrupts the connection to ICL1, but does not destabilize it?

The results for tβ1AR L72A mutant show no effect upon either agonist or antagonist binding. However, the relative deuterium uptake for unbound state of tβ1AR L72A mutant is significantly higher compared to that of the unbound of tβ1AR. This observation indicates that L72A mutation disrupts the connection between H8 and ICL1 by destabilising the ICL1itself. This finding highlights the potential role of L72 in maintaining the structural integrity of ICL1 and its connection to H8. Further details, including uptake plots for both experiments, are provided in the supplementary materials (Supplementary Figures 17-22).

The manuscript reads now:

“Interestingly, our data for the tβ1AR L72A mutant demonstrated no observable difference in deuterium uptake between the bound and unbound state of the mutant on ICL1, irrespective of whether an agonist or antagonist was present (Fig. 5H, I). However, the relative deuterium uptake for unbound state of tβ1AR L72A mutant is significantly higher compared to that of the unbound of tβ1AR. This observation indicates that L72A mutation disrupts the connection between H8 and ICL1 by destabilising the ICL1itself (Supplementary figures 17-22).”

11. Fig. 5 C-E. The illustrations could be improved by placing the labels with thin arrows next to the cartoon and perhaps using thinner helices to focus more on the structural features that these illustrations want to emphasize.

The figure has been edited according to the reviewer’s suggestion.

12. We further observed protection on ICL1 and H8, which we hypothesize contributes to the stabilization of the tβ1AR/miniGs complex by “locking” the receptor in a stable conformation.” Are there also alternative interpretations of the data? Can

the authors completely rule out that the ICL1 and H8 have a role during MiniG binding by being part of the interface?

While our hypothesis suggests that the observed protection on ICL1 and H8 contributes to stabilising the $t\beta 1AR$ /miniGs complex by "locking" the receptor in a stable conformation, we acknowledge that there could be alternative explanations. Indeed, it is plausible that ICL1 and H8 may also have a role during miniGs binding by directly participating in the interface between the receptor and miniGs. Our data provide evidence of protection on ICL1 and H8, which could indicate their involvement in stabilising the receptor-G protein complex. However, we cannot completely rule out the possibility that these regions serve as part of the interface for miniGs binding. This has been now noted in the discussion paragraph of the manuscript, which it now reads:

"We further observed protection on ICL1 and H8, which we hypothesize contributes to the stabilisation of the $t\beta 1AR$ /miniGs complex by "locking" the receptor in a stable conformation. Nevertheless, the potential involvement of ICL1 and H8 in the coupling interface cannot be excluded (Fig. 6A)."

13. "We hypothesize that the compromised protection on ICL1 and H8 in the $t\beta 1AR$ L72A mutant may be indicative of structural rearrangements that prevent the receptor from adopting a stable conformation that allows for interactions with H5 of miniGs." See last comment. Can the authors rule out that productive coupling to the G protein requires direct contact with ICL1 and H8?

While our hypothesis suggests that the compromised protection observed on ICL1 and H8 in the $t\beta 1AR$ L72A mutant might signify structural alterations hindering the receptor's adoption of a stable conformation favourable for interactions with H5 of miniGs, we recognize the existence of alternative interpretations. It remains plausible that effective coupling to the G protein may necessitate direct engagement with ICL1 and H8. Our data, however, do not definitively exclude this possibility. The compromised protection noted in the $t\beta 1AR$ L72A mutant could potentially signify shifts in the conformational dynamics of these regions, thereby impacting their capacity to interact with the G protein.

We have included that statement in the manuscript, which now reads:

"Our hypothesis suggests that the alterations observed in ICL1 and H8 of the $t\beta 1AR$ L72A mutant could impede the receptor's ability to adopt a stable conformation necessary for interacting with H5 of miniGs. However, we cannot discount the possibility that effective coupling to the G protein may involve direct engagement with ICL1 and H8."

14. It would be convenient for the reader to introduce the nomenclature used for "LysX and AspZ".

According to publication referenced below the nomenclature for mentioned residues is now included in the manuscript that reads as follow:

"Studies performed on $\beta 2AR$ report that conformational changes on H5 of the $G\alpha$ subunit affect the nucleotide binding region by directly impacting on S5 and HG helix,

which interacts with guanine ring of GDP via two residues (Lys275 and Asp277)⁴⁸. Increased disorder of those residues as well as increased mobility of HG were observed in GPCR – mediated activation⁴⁸. Additionally, one of these residues (Lys275) located on HG is a part of important salt bridge formed between HG and P-loop (Glu49)”

Sun X, Singh S, Blumer KJ, Bowman GR. Simulation of spontaneous G protein activation reveals a new intermediate driving GDP unbinding. *Elife*. 2018 Oct 5;7:e38465. doi: 10.7554/eLife.38465.

REVIEWERS' COMMENTS

Reviewer #1 (Remarks to the Author):

The authors have addressed the comments reasonably well, and also mentioned why some of the suggested experiments are not feasible at their end. The revised manuscript is suitable for publication.

Reviewer #2 (Remarks to the Author):

The authors have addressed all my comments in this revised manuscript and I recommend publication.

Reviewer #3 (Remarks to the Author):

All my concerns were taken into account.